# Image Restoration via Low-Illumination to Normal-Illumination Networks Based on Retinex Theory

**DOI:** 10.3390/s23208442

**Published:** 2023-10-13

**Authors:** Chaoran Wen, Ting Nie, Mingxuan Li, Xiaofeng Wang, Liang Huang

**Affiliations:** 1Changchun Institute of Optics, Fine Mechanics and Physics, Chinese Academy of Sciences, Changchun 130033, China; 2University of Chinese Academy of Sciences, Beijing 100049, China

**Keywords:** Retinex theory, low-illumination, decomposition network, reconstruction network, enhancement network, frequency information

## Abstract

Under low-illumination conditions, the quality of the images collected by the sensor is significantly impacted, and the images have visual problems such as noise, artifacts, and brightness reduction. Therefore, this paper proposes an effective network based on Retinex for low-illumination image enhancement. Inspired by Retinex theory, images are decomposed into two parts in the decomposition network, and sent to the sub-network for processing. The reconstruction network constructs global and local residual convolution blocks to denoize the reflection component. The enhancement network uses frequency information, combined with attention mechanism and residual density network to enhance contrast and improve the details of the illumination component. A large number of experiments on public datasets show that our method is superior to existing methods in both quantitative and visual aspects.

## 1. Introduction

With the advancement of technology, people now have access to a vast amount of information from various sources, and the visual system is the main way. Images are the most intuitive and commonly used source of information in the visual system [1]. They provide a quick and easy way for people to understand and interpret the world around them. Images contain a vast amount of information, and the quality of an image can greatly impact the amount of information that can be obtained from it. However, when the illumination conditions are poor or insufficient, it has a significant impact on the quality of the image captured by the sensor. In such situations, images suffer from reduced contrast, loss of detail, high levels of noise, and low overall brightness, this can make it difficult to extract meaningful information from the image. Poor image quality can greatly impact the effectiveness of post-processing [2,3]. Therefore, for the processing of low-illumination images, digital image processing technology can be used to develop various enhancement algorithms that can improve the quality of low-illumination images. From the perspective of image processing, researchers propose many enhancement algorithms to solve the image quality problems [4,5].

In recent decades, many scholars have researched the enhancement of low-illumination images, and there have been significant advancements in low-illumination image enhancement techniques. The traditional methods are mainly based on histogram equalization(HE) [6] and Retinex theory [7]. Histogram equalization is a popular technique that can improve contrast by spreading the intensity levels of an image over the entire available range of values. This results in a more balanced distribution of brightness and contrast in the output image. The goal of Retinex-based methods is to enhance the reflection image by improving the influence of the brightness image on it. This is achieved by carefully adjusting the relationship between the two components.

In recent years, deep learning has gained significant popularity in the field of low-light image enhancement [8,9]. This is primarily attributed to its remarkable ability to capture powerful feature representations and perform nonlinear mappings. Among them, Lore et al. [10] are the first to use deep learning in the field of low-light image enhancement. The proposed method can adaptively enhance image brightness without oversaturation. In addition, some researchers have explored the combination of traditional methods and deep learning to enhance low-illumination images [11]. Among them, the most representative is the Retinex-based deep learning method [12,13]. These methods excel in enhancing the brightness of low-illumination images by improving the estimation of both the reflection component and the illumination component. Subsequent researchers have made advancements in addressing the limitations, overfitting, and real-time problems associated with deep learning methods in low-light image enhancement. These advancements improve the practicality and applicability of deep learning techniques in this field.

While deep learning-based methods have made progress in low-illumination image enhancement, there are still challenges and limitations that need to be addressed [14,15]. For instance, the enhancement effect of the fully supervised learning method is largely affected by its training data set, so there are some common problems in the enhancement effect. Consequently, there are common issues that can arise in the enhancement results, including loss of image details, amplification of noise, and color distortion [16,17]. Hence, there is a need for a more comprehensive approach that takes into account multiple factors and provides a more accurate and realistic representation of the original scene in low-illumination image enhancement. In this paper, we propose a novel fully supervised learning algorithm for low-illumination enhancement. The algorithm consists of three parts, Decom-Net, Recon-Net, and Ehance-Net. Decom-Net aims to enhance the input low-illumination image by decomposing it into an illuminance map and a reflectivity map, following the principles of the Retinex theory. Recon-Net utilizes the decomposition result as input and utilizes the illumination component as a constraint. It employs a combination of global down-sampling operation and local residual block feature extraction to effectively suppress noise in the reflection component. Additionally, a color loss function is employed to mitigate color fading in the reflection component, so as to obtain the decomposition result with better visual quality. In Ehance-Net, spatial information and frequency information are used to improve the image contrast under the attention mechanism, and the dense residual connection network is employed to better preserve the details in the image. Our contributions can be summarized as follows:We propose an effective end-to-end network based on Retinex theory. The network decomposes the image in a data-driven manner, constructs global and local residual convolution blocks for denoizing, and combines frequency information with spatial information to improve brightness and contrast.Aiming at the color degradation in low-illumination image enhancement, we propose color loss function to suppress color degradation.We conduct an extensive set of experiments to demonstrate the efficacy of our proposed model. Experimental results serve as strong evidence for the effectiveness of our approach.

The rest of this paper is organized as follows: In Section 2, we review the traditional methods and low-illumination image enhancement methods based on deep learning. In Section 3, we present the model proposed in this paper. Section 4 introduces the loss function in detail. Section 5 gives the experimental results and evaluation, and the sixth section summarizes the full text.

## 2. Related Work

### 2.1. Traditional Methods

The traditional method is mainly based on histogram equalization(HE) and Retinex theory. On the basis of Histogram Equalization (HE), Zhu et al. [18] proposed Constrained Local Histogram Equalization (CLHE). CLHE uses a sliding window to divide the image into smaller blocks, and then performs HE on each block individually. The size of the sliding window can be adjusted to suit the specific image being processed. Compared with the original HE, local details are significantly enhanced with CLHE. Wongsritong et al. [19] proposed a method called Multipeak Histogram Equalization with Brightness Preserving (MPHEBP). The method uses local maximum values in the histogram to partition the image into sub-regions. On the basis of subgraph equalization after histogram division, the distribution range of each subgraph is redistributed according to the number of pixels in each subgraph of each histogram. The partition value is the local maximum. After equalization, it is normalized, and the whole image is multiplied by a proportional coefficient to adjust the average brightness to the average brightness of the original image. Ibrahim et al. [20] proposed a method called Brightness-Preserving Dynamic Histogram Equalization (BPDHE). This method normalizes the acquired sub-regions after histogram equalization of low-illumination images, ensuring the overall brightness of the image is preserved. Finally, the whole image is multiplied by a proportional coefficient to adjust the gain, and the brightness mean is adjusted to the brightness mean of the original image.

The Retinex theory was first proposed by Land et al. [21]. Researchers make progress in improving the Retinex algorithm and addressing some of its limitations. The single scale Retinex algorithm (SSR) [21] was proposed as an improvement over the original Retinex algorithm. The algorithm adjusts the intensity values of each layer to enhance the overall appearance of the image. Jobson et al. [22] proposed the multi-scale Retinex algorithm (MSR), which builds upon the single-scale Retinex (SSR) method. The MSR approach uses a set of filters of varying sizes to capture information at multiple scales, which allows for better preservation of both local and global contrast in images. Rahman et al. [23] proposed the Multi-Scale Retinex with Color Restoration (MSRCR) method, which builds upon the MSR algorithm by addressing the issue of color distortion that can arise when enhancing contrast in local areas of an image. In addition, some scholars introduced variational models into the Retinex algorithm and achieved good results. Kimmel et al. [24] first proposed the variational Retinex model, which transforms the illumination and reflection estimation into the optimal solution problem for the multi-objective function. By iteratively solving the objective function, the optimal illumination and reflection estimation are obtained simultaneously.

### 2.2. Learning-Based Approach

In recent years, the field of low-illumination image enhancement has witnessed significant progress owing to the rapid advancements in deep learning. Many techniques such as deep convolutional neural networks and GAN are successfully employed in the domain of low-illumination image enhancement. The earliest deep learning method proposed for enhancing low-illumination images is LLNet, introduced by Lore et al. [10]. This technique involves training a stacked sparse denoizing autoencoder using synthetic data, which enables it to perform simultaneous enhancement and denoizing of images. Wei et al. [25] proposed the Retinex-Net method, which combines Retinex theory with deep convolutional neural networks to achieve enhanced low-illumination images. Retinex-Net employs a decomposition-enhancement structure to improve the contrast of the illumination component, and BM3D is used to remove noise from the reflection component. The enhanced image is obtained by fusing the two components. MSR-Net was proposed by Shen et al. [26]. In the MSR-Net, the multi-scale Retinex module is integrated into the network architecture to capture and enhance the multi-scale information present in the input images. By incorporating the multi-scale Retinex theory into the network, the MSR-Net can effectively enhance the details and improve the visual quality of low-illumination images. Lü et al. [27] proposed the Multi-branch Low Light Enhancement Network (MBLLEN), which leverages convolutional neural networks to extract rich features from images at different levels. The output results from the multiple subnets are fused together to obtain the final enhanced image, therefore improving the overall image quality from various aspects. Jiang et al. [28] were the first to propose applying unsupervised learning to low-illumination image enhancement, which they named EnlightenGAN. This method successfully introduces unpaired training to low-illumination image enhancement for the first time, enabling adaptive enhancement of local areas in images. Lim et al. [29] proposed a novel technique called deep-stacked Laplacian restorer (DSLR) for recovering both global illumination and local details from the original input in low-light image enhancement. Guo et al. [30] proposed a lightweight network called Zero-DCE, which is capable of no-reference training and avoids the risk of overfitting in low-light image enhancement. Subsequently, Li et al. [31] introduced Zero-DCE++, an extension of the original Zero-DCE method that effectively reduces the parameter count and improves the computational efficiency of the network. Liu et al. [32] proposed the RUAS network, which leverages knowledge injection of low-light images and a lightweight priority architecture, effectively reducing the computational complexity of the network. Additionally, a structure search method is employed to enhance the efficiency of the network’s overall structure.

## 3. Method

According to Retinex theory, the perceived color and brightness of an object by the human eyes depend on the reflection characteristics of the object surface. The human visual system perceives the reflectivity of the object under different illumination conditions and maintains color constancy. This is because the human visual system takes into account both the spectral content of the illumination and the reflectance properties of the object being viewed. Based on this feature, Land et al. [21] constructed the following mathematical model:(1)S=R∘I

Among the Retinex theory, *R* represents the reflection component of an image, which reflects the essential characteristics of the object being viewed and has nothing to do with the light. *I* represents the illumination component of an image, which reflects the effect of the illumination on the object. The illumination component varies depending on the lighting conditions and can affect the perceived color and brightness of the object. ∘ denotes multiplication by elements. *S* denotes the original image.

In the case of sufficient illumination, this image is called a normal-illumination image ($S_{nor}$), and the illumination distribution inside the image will change continuously. Such an image excels at effectively preserving the image’s edges and intricate details. Nevertheless, in cases where the light source’s brightness is insufficient, the resulting image exhibits reduced overall luminosity, termed as a low-illumination image ($S_{low}$), often resulting in the loss of image detail information. Because photographic equipment cannot capture the reflected light information on the surface of an object perfectly, images often contain a large amount of noise, which is mainly distributed in the reflection component. Therefore, when enhancing low-illumination images, it is necessary to consider noise suppression of the reflection component obtained in the decomposition network and enhance the illumination component to improve the image brightness and contrast.

The network framework design of this paper is shown in Figure 1. The overall framework of the network is divided into three parts: Decom-Net, Recon-Net, and Enhance-Net. Initially, Decom-Net serves as the input layer for the entire network. During the training phase, both the low-illumination image ($S_{low}$) and the normal-illumination image ($S_{nor}$) from the paired dataset [25] are used as inputs. However, only the low-illumination image (Slow) is needed during the testing phase. The low-illumination image ($S_{low}$) and the normal-illumination image ($S_{nor}$) in Decom-Net share the same network parameters. Following the principles of the Retinex theory, the image is decomposed into the reflection component image ($R_{low/nor}$) and the illumination component image ($I_{low/nor}$). Subsequently, the reflection component ($R_{low}$) is merged with the illumination component ($I_{low}$) to form the input for Recon-Net. Recon-Net employs both global and local residuals to effectively reduce noise in the reflectance map. The frequency information from the illumination component ($I_{low}$) acquired through Decom-Net, and the reflection component ($R_{rec}$) obtained from Recon-Net are utilized as inputs for Enhance-Net. In Ehance-Net, spatial information and frequency information are combined, an attention mechanism is used to highlight useful information, and a residual dense network is used to enhance detailed information. Finally, the reflection component ($R_{rec}$) and the illumination component ($I_{en}$) are acquired through Recon-Net and Enhance-Net, followed by the generation of an enhanced image with natural brightness through pixel-by-pixel multiplication.

### 3.1. Decom-Net

The key to Retinex-based methods is to obtain a high-quality illumination map and reflectance map; the quality of the decomposition results also affects the subsequent enhancement and denoizing processes. The approach proposed in this paper utilizes data-driven learning to enhance the universality of the network. As shown in Figure 2, the low-illumination image ($S_{low}$) and the normal-illumination image ($S_{nor}$) are used as the input of the Decom-Net, and they are decomposed into the reflection component ($R_{low/nor}$) and the illumination component ($I_{low/nor}$).

In contrast to series convolutional layers employed in Retinex-Net [25] that primarily emphasizes high-level semantic information, our approach incorporates an encoder-decoder network with hierarchical feature mapping to capture multiscale context information. By incorporating skip connections, our method not only helps alleviate overfitting, but also facilitates the integration of low-level visual features with high-level semantic information.

To begin with, a 9 × 9 convolutional layer is applied to the input image to extract its low-level features. The resulting feature maps are then fed into two separate branches for further processing. The first branch includes 4 convolutional layers of 3 × 3. This branch is designed to convert low-dimensional features into high-dimensional ones. The second branch employs a shallow up-down sampling structure to process the global information of the image. To address the issue of lost feature information caused by the lack of translation invariance in the max pooling layer, we opt to use a 2 × 2 convolutional layer with a stride of 2 for down-sampling instead. And to reduce the loss of features during this process, the skip connection, which preserves the global context information, is introduced after the image up-sampling.

Finally, to ensure that the reflection and illumination components are within the range of [0, 1], the Sigmoid function is applied as a constraint. Additionally, in order to reduce training time, weight sharing is introduced in the Decom-Net.

### 3.2. Recon-Net

Low-illumination images often contain a significant amount of noise in dark areas due to factors such as shooting equipment and ambient light sources. The distribution of noise on the reflection component is complex and highly dependent on the brightness distribution of the illumination component. Unprocessed noise can significantly degrade the quality of an enhanced image. To address this issue and better preserve details in the reflected component ($R_{low}$) image, we design the reconstruction network which removes noise and achieves our desired outcome.

The Recon-Net is shown in Figure 3. The network consists of an encoding network and a corresponding decoding network. Both the illumination component and reflection component are used as input. Specifically, the spatial features are down-sampled to decrease the feature dimension and achieve an effective feature representation. While our approach is effective at removing pixels containing noise, it also leads to image blurring. To address this issue, a skip connection, which compensates for the loss of detailed information, is introduced during down-sampling. By introducing this information during up-sampling and restoring the original image size, which preserves image sharpness and improves overall image quality.

Figure 4 illustrates the structure of encoding and decoding network units, which are comprised of residual units. The encoding network unit is made up of several components, including 3 × 3 convolutional layers, 1 × 1 convolutional layer, dilated convolution, and identity mapping. Specifically, when a feature map is input into an encoding network unit, two convolutions are applied to extract its features. Then the original feature map and the extracted feature map are added by identity mapping, which helps to improve the overall feature learning ability of the network. To increase the receptive field of our encoding network unit without sacrificing resolution, the network uses dilated convolution. This allows the network to capture more context information while avoiding the need for additional network depth. After dilated convolution, the original feature map is added to the extracted feature map features. Our decoding network unit is similar to our encoding network unit, consisting of 3 × 3 convolutional layers. The use of identity mapping promotes the flow of shallow features to deep features, helping to reduce the loss of valuable feature map information and enhance model performance.

The reconstruction network takes in both the reflection and illumination components as inputs. The encoding and decoding network extracts features from these components. To restore the image size after down-sampling and to prevent image distortion during decoding, the sub-pixel convolutional layer is employed in the decoding network to restore the image size. Finally, the reflection component ($R_{low}$) is reconstructed by decoding network units.

### 3.3. Enhance-Net

The illumination component obtained from the Decom-Net may have low brightness and poor detail. To enhance both the brightness and detail of this component in low-light images, further processing is required. Therefore, combined with the detail information of the reflection component, enhancement network is designed to solve the problem.

The structure of the brightness adjustment network is illustrated in Figure 5. The network uses the frequency information of the illumination component of the decomposition network ($I_{low-fre}$) and the output of the reconstruction network ($R_{rec}$) as input to enhance the brightness of $I_{low}$. To obtain the final enhanced image output, the enhancement illumination component ($I_{en}$) and the reconstruction component ($R_{rec}$) are combined. This combination is used to enhance the overall image quality and produce the output image ($S_{en}$).

The module consists of two sub-networks. The first sub-network is the encoding and decoding network with attention module (EDNA). To improve the illumination adjustment process, an attention mechanism module is incorporated into the Enhance-Net. This module reduces the feature response to irrelevant background information while enhancing the processing ability of the algorithm for detail features. As illustrated in Figure 6, the input to the attention mechanism module comprises the image features ($x_{i}$/$y_{i}$) obtained from the lower and upper sampling layers, respectively. After processing by the attention module, the response to brightness features is enhanced, allowing the output feature ($z_{i}$) to carry more brightness information.

To process the inputs $x_{i}$ and $y_{i}$ in the attention mechanism module, each is passed through an independent 1 × 1 convolutional layer. The resulting features are then added together before being activated by the LeakyReLU activation function. After passing through the 1 × 1 convolutional layer and the Sigmoid function, the resulting features are interpolated using bicubic interpolation to restore the original size of the feature map. The final output feature ($z_{i}$) is obtained by multiplying the interpolated feature map with $x_{i}$. The attention mechanism in this propagation process can effectively fuse image information from different scales, reduce the response of irrelevant features, and enhance the network’s ability to adjust brightness.

The second sub-network of the Enhance-Net consists of a residual dense layer comprised of residual dense blocks (RDB); the structure is shown in Figure 7. The connection structure used in the residual dense blocks (RDB) allows the features from all previous convolutional layers to be combined before transmitting the features backward through the network. This leads to the formation of a contiguous memory (CM) mechanism, which helps to improve the learning and utilization of feature information during the illumination adjustment process. To address issues such as overfitting and gradient disappearance, the result from the feature extraction from the previous layer is used as the input to each residual block. Each residual block within the RDB comprises three 3 × 3 convolutional blocks, a 1 × 1 convolution, and LeakyReLU activation function. After several convolutional and LeakyReLU activation function operations, the extracted feature information is concatenated and added to the input of the current residual block using an addition operation. The output of each residual block is then passed as input to the next layer.

The high-frequency information of the image has more and more obvious brightness changes, and the low-frequency information represents a general overview and contour. Therefore, it is combined with the reflection component ($R_{rec}$) as the network input.the output structures of the two sub-networks are concatenated, the enhancement illumination component ($I_{en}$) is obtained by merging 1 × 1 convolutional layer.

### 3.4. Loss Function

To train the network effectively, an appropriate loss function must be established to estimate the weights of the network and improve the accuracy of learning. This loss function should take into account the desired properties of the output image, such as brightness and detail enhancement, while also minimizing artifacts and noise in the image.

#### 3.4.1. Decom-Net

The Decom-Net is designed to tackle the task of separating the illumination and reflection components from input low-illumination images. The loss function of the Decom-Net consists of four components:(2)$$L_{Decom}=L_{rec}+\lambda_{r}L_{R}+\lambda_{is}L_{is}+\lambda_{vgg}L_{vgg}$$

Among them, $\lambda_{r}$, $\lambda_{is}$, and $\lambda_{vgg}$ are the weight coefficients of consistency loss, illuminance smoothness, and perceptual loss.

Retinex theory suggests that the reflection component of an image should preserve its intricate texture features, while the illumination component ought to exhibit a smooth and uniform appearance. As a result, the reconstruction loss function of the constrained Decom-Net takes the following form:(3)$$L_{rec}=∥R_{low}\circ I{}_{nor}-{S_{low}∥}_{1}+∥R_{nor}\circ I{}_{low}-{S_{nor}∥}_{1}$$

Using the Decom-Net to output the results of different illuminations for pixel multiplication, the results of Decom-Net are closer to the real image effect.

Retinex theory’s color constancy dictates that the reflection component ($R_{nor/low}$) of a paired low-illumination image ($S_{low}$) and normal-illumination image ($S_{nor}$) should exhibit maximum consistency. To achieve this, consistency loss is introduced for reflection components, as per the following formula: (4)$$L_{R}=∥{R_{nor}-R_{low}∥}_{1}$$

In natural settings, images are expected to have smooth illumination. Hence, it becomes crucial to limit the horizontal and vertical gradients of every pixel in the image to avoid excessively large variations. The loss function uses the gradient term of the reflection component as the gradient map of the illumination component to assign weights, so as to better preserve the texture details and boundary information in the brightness smoothness of the illumination map. The loss function is as follows:(5)$$L_{is}=\sum_{n=low,normal}∥{\nabla I_{n}•\exp\left(-\lambda_{s}\nabla R_{n}\right)∥}_{1}$$

∇ represents the sum of horizontal and vertical gradient in the image, while $\lambda_{s}$ refers to the perceptual gradient balance coefficient. Notably, exp($-\lambda_{s}\nabla R_{n}$) helps to ease the smoothing constraint at steeper gradients of the reflection component.

The addition of perception loss [33] to the Decomposition Network aims to align its output with the perceptual effect observed by human eyes. The corresponding loss function is as follows:(6)$$L_{vgg}=\frac{1}{C_{j}H_{j}W_{j}}\sum_{n=low,normal}{∥R_{n}\circ I_{n}-S_{n}∥}_{2}^{2}$$

Among them, $C_{j},H_{j},W_{j}$ represent the size of the feature map of the *j* layer.

#### 3.4.2. Recon-Net

The Reconstruction Network is designed to retain the texture detail information of the reflection component while simultaneously suppressing noise. However, during the up- and down-sampling process, images may lose details and experience color fading. Therefore, a well-designed loss function is necessary to constrain the final enhancement outcome and guide the network toward convergence in the right direction. The following loss function is employed for this purpose in the Recon-Net:(7)$$L_{Recon}=\lambda_{m}L_{Mix}+\lambda_{col}L_{color}+\lambda_{tv}L_{tv}$$

In this context, $\lambda_{m}$, $\lambda_{col}$ and $\lambda_{tv}$ denote the weighting coefficients for consistency loss, color loss, and total variation loss, respectively.

To enhance the detail expression ability of the Reconstruction Network and ensure consistency in structure between the enhanced reflection component ($R_{rec}$) and the decomposition network’s decomposed reflection component ($R_{nor}$), we use a loss function, such as:(8)$$L_{Mix}=\alpha •L_{MS-SSIM}+\left(1-\alpha \right)•G_{\sigma_{G}^{M}}•L_{l_{1}}$$

Although MS-SSIM may cause changes in brightness and color accuracy, it is effective in retaining the edge details of the image. The $L_{l_{1}}$ loss function can maintain the invariance of brightness and color. By combining these two loss functions using a weighted sum, the mixing loss aims to balance the importance of structural information with the need to minimize the absolute error.

Among them, $L_{MS-SSIM}=1-\left[MS-SSIM\left(R_{nor},R_{rec}\right)\right]$, $L_{l_{1}}={∥R_{nor}-R_{rec}∥}_{1}$.

In this context, the parameter G represents the Gaussian distribution, while α is a constant that has been set to 0.84. The mixing loss function is a combination of two other loss functions: MS-SSIM and $L_{l_{1}}$. The weights assigned to each of these loss functions are determined by the value of α.

The $R_{low}$ contains color information for the low-illumination scenario, and it is expected to have consistent color information with $R_{nor}$. Therefore, using $R_{nor}$ as the reference image, the color loss function is designed to make $R_{low}$ restore the normal color information:(9)$$L_{color}=\frac{1}{C_{j}H_{j}W_{j}}\sum_{i=R,G,B}sim(R_{nor}^{i},R_{low}^{i})$$

The cosine similarity [34] is more to distinguish the difference from the direction, and is not sensitive to the absolute value, so it is impossible to measure the difference in the value of each dimension. Therefore, the color loss is reduced by adjusting the cosine similarity. sim(·,·) represents the adjusted cosine similarity. The calculation of the adjusted discrete cosine similarity between the fused image and the original visible image in the R, G, and B channels.

Rudin et al. [35] pointed out that images affected by noise typically have a higher total variation compared to noise-free images. By limiting the total variation, it is possible to suppress the effect of noise in the image. To promote spatial smoothness of network output results and remove noise, a TV (Total Variation) Loss is used as a regularization term to constrain network learning in the population function. The loss function is formulated as follows:(10)$$L_{tv}={∥\nabla f(x)∥}_{1}$$

∇*f*(*x*) represents the gradient of the output image *f*(*x*).

By minimizing the TV Loss, the model is encouraged to produce output images with smoother transitions between pixel values, which can help reduce the amount of noise present while preserving important features and details in the image.

#### 3.4.3. Enhance-Net

To capture more comprehensive brightness information and achieve a brightness enhancement effect that closely resembles that of a normally illuminated image, we have designed the following loss function:(11)$$L_{Enhance}=L_{relight}+\lambda_{i}L_{i}$$

$\lambda_{i}$ is the weight coefficient of light loss.

Based on the Retinex theory, the product of the two is consistent with the input image. By using a loss function, the enhancement effect of the illumination component on the low-illumination image becomes more similar to that of a normally illuminated image:(12)$$L_{relight}={∥R_{rec}\circ I_{en}-S_{nor}∥}_{1}$$

To constrain the illumination component, the input image guides the illumination component, and there are significant changes in illumination at strong edges of the input image. This means that in regions where the illumination change in the input image is small, the corresponding illumination change in the enhanced image should also be small. The loss function is formulated as follows:(13)$$L_{i}=\sum_{i=en,nor}{∥\frac{\nabla I_{n}}{\max(\left|\nabla S_{n}\right|,E)}∥}_{1}$$

$I_{n}$ represents the illumination component of the enhanced/normal-illumination image, and $S_{n}$ represents the enhanced/normal-illumination image. The symbol ∇ represents the sum of the first derivatives in both horizontal and vertical directions. To avoid a zero denominator, we introduce a small positive constant *E* (set to 0.01 in this study).

## 4. Experimental Results and Analysis

The algorithm presented in this paper leverages the training set from the LOL dataset [25] and a synthetic datase [25] to effectively train the network and achieve model fitting. To assess the model’s fitting performance, evaluation metrics are applied to the test set result evaluation of the LOL dataset [25] to validate its effectiveness in achieving model fitting. Then, in order to ascertain the model’s universality, DICM dataset [36], MEF dataset [37], NPE dataset [38], and LIME dataset [39] are employed, accompanied by various evaluation metrics to assess its applicability. The training and testing experiments of the network are completed on the Nvidia RTX A6000 GPU device, and the implementation code is based on the Pytorch framework. During the training process, the network uses image pairs for training, the Batch Size is set to 32, and the Patch Size is set to 96 × 96. The methods selected for comparison include MSRCR [23], DRBN [40], RRDNet [41], Zero-DCE++ [31], Retinex-Net [25], DSLR [29], RUAS [32], URetinex-Net [12], and the results of all comparison methods are reproduced from their official code.

In all experiments, after numerous rounds of training and validation, we empirically set $\lambda_{r}$ = 0.1, $\lambda_{is}$ = 0.01 and $\lambda_{vgg}$ = 0.05 in Decom-Net. $\lambda_{m}\;=\;0.0001,\lambda_{col}\;=\;0.03$ and $\lambda_{tv}\;=\;0.0005$ in Recon-Net. Then, $\lambda_{i}$ = 0.1 in Enhance-Net. After experimental testing, the number of parameters of the Decom-Net is 1.24 M, the number of parameters of the Recon-Net is 5.34 M, and the number of parameters of the Enhance-Net is 0.43 M. The parameter of the total network is 19.92 M.

### 4.1. Ablation Experiment

To investigate the effectiveness of different components in the model and the loss function settings, we conduct a quantitative evaluation of our network using the LOL dataset. We use PSNR to measure the noise level, and SSIM to evaluate overall image quality, including brightness, contrast, and structure. The results of this evaluation are summarized in Table 1.

Numbered lists can be added as follows:To assess the effectiveness of the encoding and decoding network unit (En/De-Unit) in Recon-Net, we replaced it with four 3 × 3 convolutional layers to form a standard unit (Conv-Unit). The PSNR and SSIM values obtained using the standard unit were 17.157 and 0.704, respectively, indicating that the encoding and decoding network unit performs better than the standard unit. Experimental results demonstrate that our architecture yields better performance overall.The Recon-Net and Enhance-Net networks have been integrated into a single network, and the loss function is jointly trained, which has verified the necessity of the existence of the two networks. The experiments reveal a notable decrease in both PSNR and SSIM indicators. This outcome serves as evidence for the indispensability of the two subnetworks’ existence.To evaluate the effectiveness of the two sub-networks in Ehance-Net, we removed them and tested the modified network. The experimental results show that combining the two sub-networks can achieve better performance than using either network alone.To validate the effectiveness of our loss function, we conducted experiments by removing the perceptual loss and replacing the $L_{l_{1}}$-MS-SSIM mixed loss function with an $L_{l_{1}}$ loss function. After removing the perceptual loss, the PSNR was 17.324 and SSIM was 0.719. After replacing the $L_{l_{1}}$-MS-SSIM mixed loss function, the PSNR was 17.460 and SSIM was 0.721. The removal or replacement of the loss function resulted in a degradation of network performance. These experimental results provide strong evidence for the rationality of our loss function setting.

### 4.2. Subjective Evaluation

We show the enhancement effects of various algorithms in figures obtained from DICM [36] and LOL [25] datasets, respectively, showcasing the enhancement effects of various algorithms. Some details are enlarged to facilitate subjective visual evaluation. Among these Figure 8, Figure 9, Figure 10 and Figure 11, MSRCR shows a better visual brightness improvement effect, but the overall clarity of the enhanced image is lacking, and there are some defects in color retention. DRBN effectively enhances image details, but the improvement in image brightness is somewhat limited, and the color restoration effect is poor. RRDNet performs well on brighter original images, but its performance is relatively poor on darker images with no significant brightness improvement, making it difficult to achieve satisfactory visual effects. While DSLR provides some improvement in the brightness enhancement effect of certain areas in the original image, there is a noticeable stacking occlusion in the enhancement effect. Zero-DCE++ can effectively preserve the detailed features of an image, but its brightness enhancement effect is not very pronounced. Additionally, the color contrast of the image is significantly reduced. The Retinex-Net greatly enhances the brightness; however, it also leads to color fading and significant noise in certain areas. However, it is prone to producing artifacts in the enhanced image. The RUAS algorithm exhibits superior image enhancement results in regions with gradual illumination changes. However, it may lead to overexposure in areas characterized by significant variations in illumination. While the URetinex-Net method has shown promising results in terms of color consistency and detail enhancement. However, in the environment with a strong illumination background, there will be overexposure problems, resulting in the loss of some details in the region. Compared to other methods, our approach effectively avoids common issues such as color distortion, detail loss, and artifacts, while improving the brightness of the original image overall without underexposure. Although our method may produce overexposure in regions with high brightness in the original image, this is a minor drawback in comparison.

In Figure 8, Figure 9, Figure 10 and Figure 11. Some areas in the picture are marked with red frames and enlarged. Based on the magnified area, it is evident that the algorithm proposed in this paper excels in preserving details, restoring colors, and enhancing brightness.

### 4.3. Objective Evaluation

In terms of objective evaluation, this paper utilizes four indicators to assess the performance of the algorithm: PSNR (Peak Signal-to-Noise Ratio), SSIM (Structural Similarity Index Measure), NIQE (Natural Image Quality Evaluator), and LOE (Lightness Order Error). PSNR is a widely used index for measuring the effectiveness of image enhancement algorithms. It indicates the ratio of the peak signal in an image to the noise signal. Generally, a higher PSNR value indicates a better image enhancement effect. SSIM is a metric that is not affected by changes in brightness or contrast, and it reflects the degree of similarity between the enhanced image and the original input image. The SSIM value ranges between 0 and 1, with higher values indicating greater similarity between the processed image and the original input image. NIQE is a metric that is based on the construction of a series of features for measuring image quality, which are then used to fit a multivariate Gaussian model. The smaller the value, the better the visual effect. LOE reflects the naturalness of the enhanced image. For LOE, the smaller the value, the better the brightness order is preserved. The evaluation of an image using this model essentially measures the difference in the multivariate distribution of the image being tested. As shown in Table 2, the proposed method ranks second and first in PSNR and SSIM metrics. In tests conducted on DICM, LIME, MEF, and NPE datasets, our method has achieved first and third results in NIQE and LOE indicators. Combining these indicators, the performance of this algorithm is improved compared with other algorithms. Table 2, Table 3 and Table 4 show the effectiveness and applicability of the proposed method.

In Table 5, we utilize the data set VE-LOL-H [42] to compare the running times of various algorithms and test them on the NVIDIA RTX A6000 device. When compared with MSRCR, RRDNet, and DRBN, this algorithm demonstrates superior performance on high-resolution images. In comparison to Zero-DCE++, Retinex-Net, RUAS, and DSLR, although this algorithm is slower in running speed, it takes into account the impact of noise, texture details, and color fading comprehensively, resulting in superior processing results.

## 5. Conclusions

In this paper, we propose a low-illumination image enhancement algorithm based on the Retinex theory, which consists of three sub-networks: Decom-Net, Recon-Net, and Enhance-Net. Our proposed method yields enhanced images that exhibit superior visual quality compared to those produced by existing methods. Experimental results on public datasets demonstrate that our approach can effectively improve image contrast while suppressing noise more effectively than other methods. Overall, our proposed method outperforms existing techniques for low-illumination image enhancement.

In general, the main contributions of this paper are as follows:In this study, we propose a novel end-to-end neural network aimed at converting low-illumination images into normal-illumination images. In order to solve the problem of noise and low brightness in low-illumination images, we combine global and local residual connections, and use spatial information and frequency information to design a new network structure.To constrain the neural networks, we propose and introduce several loss functions. By leveraging these loss functions, we are able to enhance image brightness, suppress noise and color fading issues, and more effectively preserve the texture details of the image.The public experimental results show that our method has improved most of the indicators compared with the existing technology. The enhancement results produced by our approach are better equipped to handle the noise issues that inevitably arise during the process of brightness enhancement, while also effectively fulfilling the task of improving brightness and reducing noise simultaneously.

In our future research, we plan to refine the network structure and loss function to develop a more advanced low-illumination image enhancement method. We also aim to apply this method to more complex visual tasks, such as target detection and face recognition in low-light environments.

## Figures and Tables

**Figure 1 sensors-23-08442-f001:**
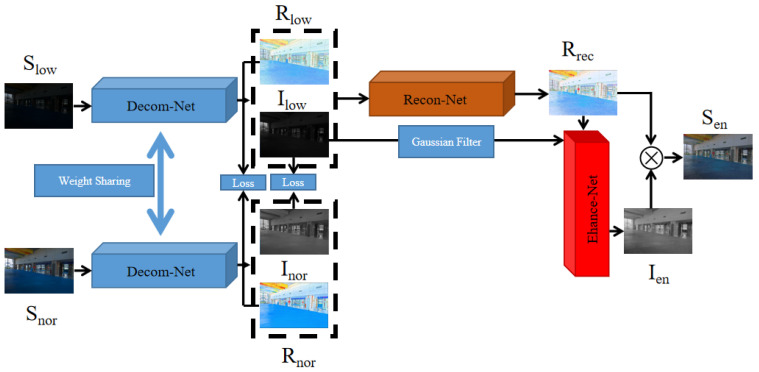
The framework of the proposed model. Our network comprises three subnets: Decom-Net, Recon-Net, and Enhance-Net.

**Figure 2 sensors-23-08442-f002:**
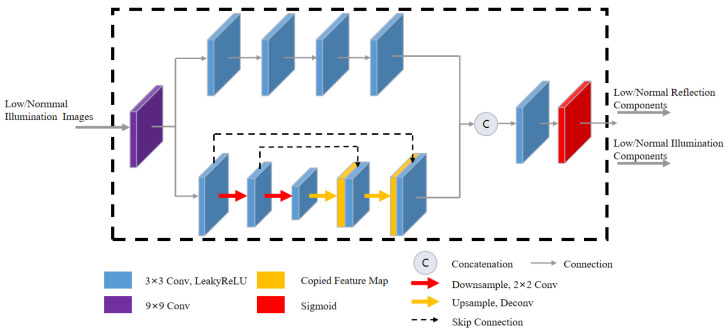
The proposed Decom-Net architecture. The Decom-Net consists of two modules: The Stacked Convolution Module (SCM) and the Encoding–Decoding Module (EDM). SCM primarily emphasizes high-level semantic information. EDM focuses on global information and enhances the flow of feature information.

**Figure 3 sensors-23-08442-f003:**
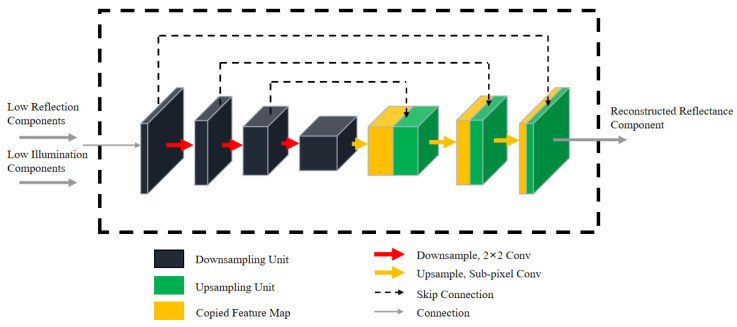
The proposed Recon-Net architecture. The network consists of encoding and decoding network units. The encoding network unit performs feature extraction to obtain abstract features, while the decoding network unit uses these abstract features to recover the original image size.

**Figure 4 sensors-23-08442-f004:**
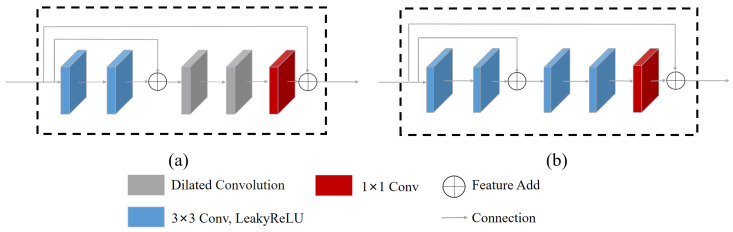
The proposed encoding and decoding network units architecture. (**a**) encoding network unit (**b**) decoding network unit.

**Figure 5 sensors-23-08442-f005:**
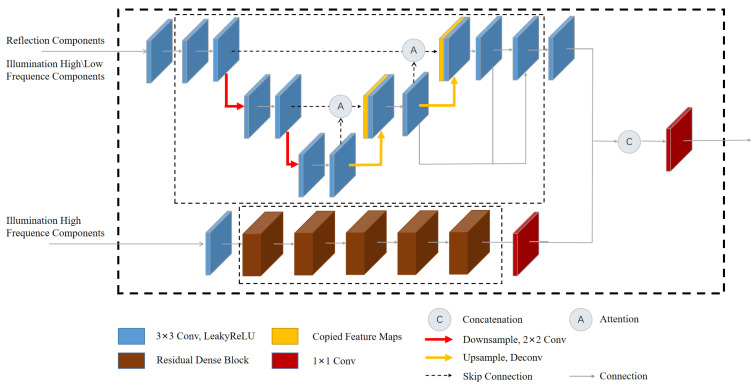
The proposed Enhance-Net architecture.The Enhance-Net consists of two modules: Encoding and Decoding Network with Attention Module (EDNA) and Residual Dense Layer (RDL). EDNA reduces the response of irrelevant features and enhances the network’s ability to adjust brightness and RDL improves the utilization of feature information.

**Figure 6 sensors-23-08442-f006:**
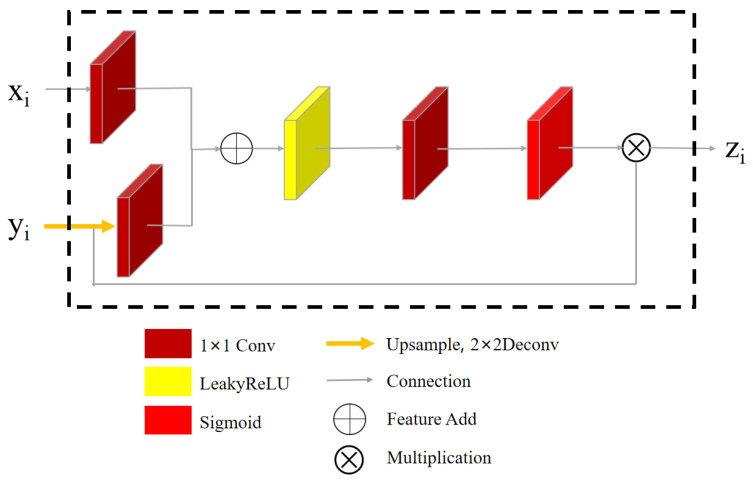
The proposed Attention Module architecture.

**Figure 7 sensors-23-08442-f007:**
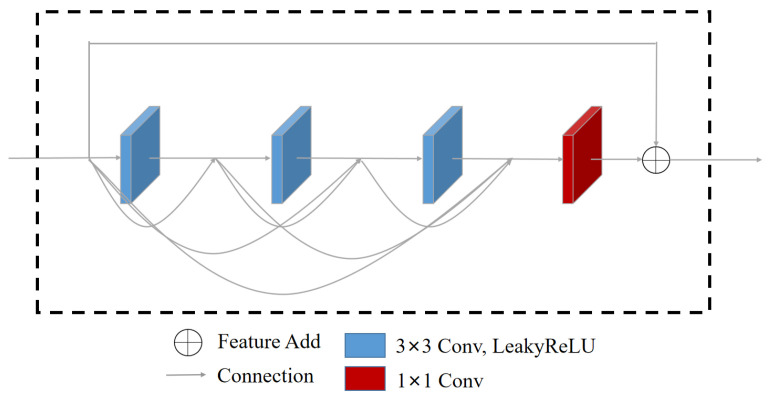
The proposed Residual Dense Block architecture.

**Figure 8 sensors-23-08442-f008:**
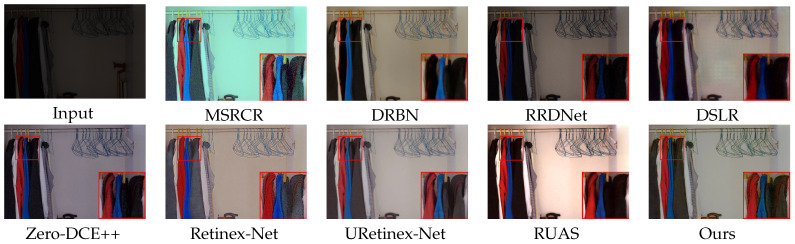
Visual comparison with state-of-the-art low-illumination image enhancement methods on the LOL dataset.

**Figure 9 sensors-23-08442-f009:**
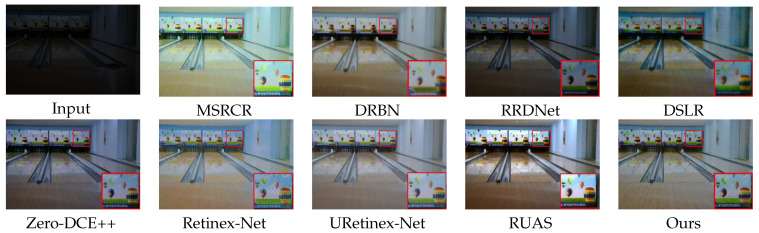
Visual comparison with state-of-the-art low-illumination image enhancement methods on the LOL dataset.

**Figure 10 sensors-23-08442-f010:**
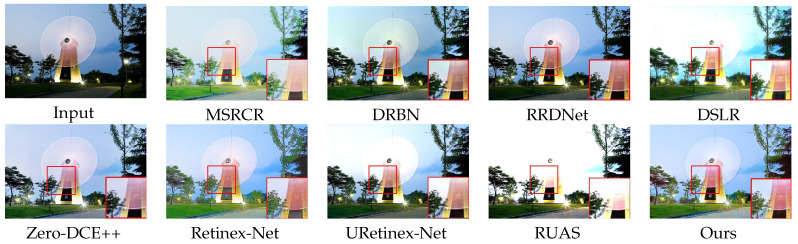
Visual comparison with state-of-the-art low-illumination image enhancement methods on the DICM dataset.

**Figure 11 sensors-23-08442-f011:**
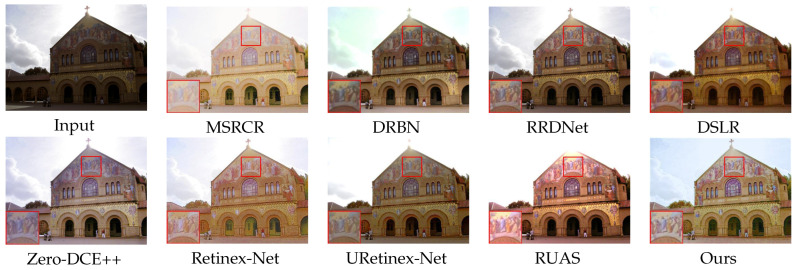
Visual comparison with state-of-the-art low-illumination image enhancement methods on the DICM dataset.

**Table 1 sensors-23-08442-t001:** Ablation experiment results of improved modules and loss.

Conditions	PSNR	SSIM
default	18.137	0.795
En/De-Unit→Conv-Unit	17.294	0.711
Recon-Net+Ehance-Net	15.315	0.571
w/o EDNA	15.581	0.627
w/o Residual Dense Net	16.791	0.673
w/o Perception Loss	17.387	0.722
$L_{l_{1}}$-MS-SSIM Loss→$L_{l_{1}}$	17.475	0.727

**Table 2 sensors-23-08442-t002:** The LOL dataset (paired dataset) was quantitatively evaluated according to PSNR and SSIM. The best results are shown in bold, the second is italic, and the third is underlined.

Method	PSNR↑	SSIM↑
MSRCR	12.342	0.423
DRBN	15.141	0.514
RRDNet	11.365	0.505
DSLR	14.976	0.612
Zero-DCE++	14.914	0.561
Retinex-Net	16.013	0.663
RUAS	17.870	0.724
URetinex-Net	**18.314**	*0.781*
Ours	*18.137*	**0.795**

**Table 3 sensors-23-08442-t003:** NIQE scores on the DICM, LIME, MEF, NPE datasets. The best results are shown in bold, the second is italic, and the third is underlined.

Method	DICM	LIME	MEF	NPE	AVG↓
MSRCR	3.878	4.462	4.312	4.552	4.301
DRBN	4.090	4.079	4.276	4.007	4.113
RRDNet	3.315	3.921	3.861	**3.754**	3.713
DSLR	4.112	4.017	4.239	3.965	4.083
Zero-DCE++	*3.094*	**3.588**	3.796	*3.838*	*3.579*
Retinex-Net	4.412	4.112	4.316	4.534	4.344
RUAS	5.217	4.026	3.834	5.534	4.653
URetinex-Net	3.386	5.518	**2.932**	6.570	4.602
Ours	**2.917**	*3.591*	*3.416*	4.071	**3.499**

**Table 4 sensors-23-08442-t004:** LOE scores on the DICM, LIME, MEF, NPE datasets. The best results are shown in bold, the second is italic, and the third is underlined.

Method	DICM	LIME	MEF	NPE	AVG↓
MSRCR	493.650	535.915	439.523	426.563	473.913
DRBN	423.548	459.371	435.135	401.619	429.918
RRDNet	**85.742**	**67.722**	**114.117**	**65.487**	**83.267**
DSLR	316.626	228.757	289.323	376.970	302.919
Zero-DCE++	377.611	306.983	385.201	325.357	348.788
Retinex-Net	457.657	621.855	444.695	451.971	494.045
RUAS	260.419	*121.855*	314.289	555.280	312.961
URetinex-Net	*216.671*	138.746	*154.144*	*225.101*	*183.666*
Ours	307.181	289.754	260.310	250.551	276.949

**Table 5 sensors-23-08442-t005:** Average runtime (RT) comparison (in seconds).

Method	Runtime
MSRCR	4.328
DRBN	0.843
RRDNet	56.365
DSLR	0.576
Zero-DCE++	0.316
Retinex-Net	0.491
RUAS	0.547
URetinex-Net	0.703
Ours	0.691

## Data Availability

The data presented in this study are openly available at https://daooshee.github.io/BMVC2018website/, accessed on 10 August 2018.

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
