# Peer review of "Image Restoration via Low-Illumination to Normal-Illumination Networks Based on Retinex Theory"

_sensors, 2023, doi:10.3390/s23208442_

Round 1
Reviewer 1 Report (Previous Reviewer 3)
The authors have significantly improved the manuscript after the revision, carefully taking into account the suggestions from the reviewers.
The ablation study on the end-to-end architecture vs the current architecture shows that the separation of architecture is required.
The experiments section contains good amount of qualitative and quantitative evaluation and proper justification.
The manuscript can be accepted in its present form.
Author Response
Thank you very much for your review of the manuscript. I revise the manuscript according to your suggestion. By enhancing the experimental design and article structure, we made the manuscript more logically coherent. Therefore, thank you very much for your suggestions for the article.

Reviewer 2 Report (New Reviewer)
This manuscript made an interesting study about image restoration about low illumination images. Authors separate lights to reflection parts and illumination parts and use unet attention mixture to enhance image. Careful comparisons are made, ablation experiments are done. From my opinion, this is also a reasonable result. This work is well organized and well written, can be accepted after making following minor consideration.
1. Inference time is quite important in industrial implementation, authors are suggested to list inference time and relevant hardware for reference. Authors made a Table 5 runtime comparison, is it an inference time? If so, authors are suggested to claim hardware used, as different structures may have large different inference time on different hardware. For example, many hardware don’t support attention structure.
2. Authors are supposed to list parameters amount for reference, as we need to know what cause the improvement, structure or increase of parameter amount.
3. Authors are suggested to improve their results as some scores are not best, in many cases, increasing channels, parameters amount can easily achieve that.
4. Have authors tried diffusion models?
Author Response
General Comments:
This manuscript made an interesting study about image restoration about low illumination images. Authors separate lights to reflection parts and illumination parts and use unet attention mixture to enhance image. Careful comparisons are made, ablation experiments are done. From my opinion, this is also a reasonable result. This work is well organized and well written, can be accepted after making following minor consideration.
Response:
We really appreciate your comprehensive comments and constructive suggestions. We have benefited greatly from your suggestions. We read and consider each comment very carefully, and thoroughly revise the manuscript according to your comments and suggestions. We hope that the manuscript reads more convincingly after the revision.
Comment 1:
Inference time is quite important in industrial implementation, authors are suggested to list inference time and relevant hardware for reference. Authors made a Table 5 runtime comparison, is it an inference time? If so, authors are suggested to claim hardware used, as different structures may have large different inference time on different hardware. For example, many hardware don’t support attention structure.
Response:
Thank you for your advice. The data in Table 5 is the inference time. The hardware I use is RTX A6000, and the inference time of each algorithm is tested on this platform. I supplement the information of the hardware platform in the article, and improved the results of the article.
Comment 2:
Authors are supposed to list parameters amount for reference, as we need to know what cause the improvement, structure or increase of parameter amount.
Response:
Thank you for your valuable suggestions regarding this article. In the experimental results and analysis section, I listed the number of parameters of the three sub-networks and the total network for readers to compare and improve the content of the manuscript.
Comment 3:
Authors are suggested to improve their results as some scores are not best, in many cases, increasing channels, parameters amount can easily achieve that.
Response:
Thank you for your suggestion. I change the number of channels of the network and adjust the parameter settings of perceptual loss, color loss and TV loss in the loss function and retrain the network. And I achieve better results than before in each index. For these results, I change in the manuscript and mark them out.
Comment 4:
Have authors tried diffusion models?
Response:
Thank you for your valuable suggestions regarding this article. About the diffusion model, I learned the relevant knowledge when I consulted the literature. However, there is no attempt to use the diffusion model in the design of the algorithm. I intend to add the diffusion model to the low-light image enhancement algorithm in future research to improve the performance of the algorithm. Thank you for your review.

This manuscript is a resubmission of an earlier submission. The following is a list of the peer review reports and author responses from that submission.
Round 1
Reviewer 1 Report
The authors present a method to improve the contrast of images in low light, using a network based on Retinex theory.
I can not judge the quality of the proposed network, since the authors have not explained the input of the initial part of the network. In the work, authors said that "the low-illumination image (S_low) and the normal-illumination image (S_nor) are used as input of the Decom-Net", but I do not understand what is S_low and S_nor. I think that is a very important information, essential to understand the paper.
From my point of view, the sentence: "Finally, the illumination component and the reflection component are multiplied to obtain the final enhanced image" have to be justified.
Authors have to explain why the same set of images is used as training set and as test set. Also, they must explain in a reasoned way the set of parameters used.
On the other hand, the paper presents some gramatical and orthographical errors which have to be improved. For example, we find many spaces missing, after periods or commas; some sentences are incomplet (71 or 277); some sentences start with lowercase; some sentences are repeated (121); ...
Author Response
General Comments:
The authors present a method to improve the contrast of images in low light, using a network based on Retinex theory.
Response:
We really appreciate your comprehensive comments and constructive suggestions. We have benefited greatly from your suggestions. We read and consider each comment very carefully, and thoroughly revise the manuscript according to your comments and suggestions. We hope that the manuscript reads more convincingly after the revision.
Comment 1:
I can not judge the quality of the proposed network, since the authors have not explained the input of the initial part of the network. In the work, authors said that "the low-illumination image (S_low) and the normal-illumination image (S_nor) are used as input of the Decom-Net", but I do not understand what is S_low and S_nor. I think that is a very important information, essential to understand the paper.
Response:
Thanks for your advice. According to the paper Deep Retinex Decomposition for Low-Light Enhancement (https://arxiv.org/pdf/1808.04560.pdf).
The initial part of the neural network is the Decom-Net, inspired by Retinex theory, to perform image decomposition into the illumination map and reflection map. The input is a low-illumination image and a corresponding normal-light image. Where S_low represents a low-illumination image, that is, an image with lower brightness. S_nor represents a normal illumination image, indicating an image with sufficient brightness.In this paper, we employ supervised learning to train the model using the relationship between low-light images and their corresponding normal-light images. The objective is to enable the model to predict images with normal brightness when tested on a separate dataset. By leveraging the annotated pairs of low-light and normal-light images during the training process, the neural network can learn to generalize and effectively enhance the brightness and overall quality of low-light images in the test set.
Comment 2:
From my point of view, the sentence: "Finally, the illumination component and the reflection component are multiplied to obtain the final enhanced image" have to be justified.
Response:
Thanks for your suggestion. According to the article Deep Retinex Decomposition for Low-Light Enhancement (https://arxiv.org/pdf/1808.04560.pdf).
I have re-expressed this sentence in the paper and cited this literature.And marked out in the text. You can see the revision (line 217).
Comment 3:
Authors have to explain why the same set of images is used as training set and as test set.
Response:
Thanks for your suggestion. According to the article Deep Retinex Decomposition for Low-Light Enhancement (https://arxiv.org/pdf/1808.04560.pdf). The proposer of the algorithm has divided the same data set into training set and test set.
Separating the training set and the test set from the data set is a practical method commonly used in deep learning. Its main purpose is to evaluate the performance of the model, select the best model and parameter configuration, and provide a reliable evaluation of the generalization ability of the model. The training set is used for data samples of model fitting to debug parameters in the network. The test set is used to evaluate the generalization ability of the final model. Dividing the data set is also to prevent model overfitting. When all the original data are used to train the model, the result is likely to be that the model fits the original data to the greatest extent, that is, the model exists to fit all the original data. When new samples appear, the model is used to predict, and the effect may not be as good as the model trained with only part of the data.
Comment 4:
Also, they must explain in a reasoned way the set of parameters used.
Response:
Thanks for your valuable suggestion. After comprehensive training on the training set, we proceed to fine-tune and optimize the model's parameters, addressing any potential issues or bugs identified during the training process. The parameter settings are iteratively refined, with certain loss functions employed by other researchers in the same field. As a result, the adjustment of parameters for these loss functions aligns with the approaches taken by fellow scholars in this domain.
Comment 5:
On the other hand, the paper presents some gramatical and orthographical errors which have to be improved. For example, we find many spaces missing, after periods or commas; some sentences are incomplet (71 or 277); some sentences start with lowercase; some sentences are repeated (121); ...
Response:
Thank you for your advice and we apologize for the grammatical errors in the paper. In response to these problems, I have made changes in the paper. The modified part has been annotated . Thank you for reviewing this article. (Line 105, line322 and line160, these lines are the modified content.)

Reviewer 2 Report
The authors proposed a deep network for low-illumination image enhancement, which was based on Retinex theory. The experiments demonstrated that the proposed method achieved comparative results in comparison with baselines. However, reviewer has following concerns:
- The novelty of the paper was not clear stated. It seems that the focus of this work is to design a dedicated Retinex-based deep model to deal with noise and color. But how these objectives can be achieved by proposed model, is not clearly mentioned in the introduction section.
- In addition, even though the authors mentioned the issues relates noise and color imbalance, the problem statement was not clearly stated. Thus, It is difficult to audience to capture the main motivation of this work.
- Since this work relies on a elaborately designed architecture, reviewer concerns about computational burden. Reviewer did not find such the evaluation in the manuscript. The authors should do comparison not only with baselines listed in the manuscript but also with some light-weight models, for example, RUAs (2021).
- Moreover, the list of references used in this manuscript is not up-to-date one, whose the years of publish are mostly before 2021. There are recently (2022-2023) works have quite similar objectives.
- The authors should put extra care for writing manuscript. In fact, there are repeated parts. For example, from line 36-47, the authors repeatedly mentioned how existing method ignore noise and color imbalance. Or the beginning character of a sentence was not capitalized. For example, line 32. Or no space between two sentence. For example, line 35.
n/a
Author Response
General Comments:
The authors proposed a deep network for low-illumination image enhancement, which was based on Retinex theory. The experiments demonstrated that the proposed method achieved comparative results in comparison with baselines. However, reviewer has following concerns:
Response:
Thank you very much for the time and effort that they have put into reviewing the previous version of the manuscript. These suggestions have enabled us to improve our work. We are very grateful for your comments on the manuscript. According to your advice, we amended the relevant part of the manuscript. All of your questions were answered one by one. We hope that the manuscript reads more convincingly after the revision.
General Comments:
The novelty of the paper was not clear stated. It seems that the focus of this work is to design a dedicated Retinex-based deep model to deal with noise and color. But how these objectives can be achieved by proposed model, is not clearly mentioned in the introduction section.
Response:
Thanks for your valuable suggestion, which can increase the readability of our article. I have rephrased the issue of not specifying the proposed model in the introduction section of the paper. The modified part has been annotated.
General Comments:
In addition, even though the authors mentioned the issues relates noise and color imbalance, the problem statement was not clearly stated. Thus, It is difficult to audience to capture the main motivation of this work.
Response:
Thanks for your suggestion. As for the problem that the model is not explicitly mentioned in the introduction, I have re-stated it in the introduction of the paper. And marked out in the text.
General Comments:
Since this work relies on a elaborately designed architecture, reviewer concerns about computational burden. Reviewer did not find such the evaluation in the manuscript.
Response:
Thank you for this valuable suggestion. For the problem of computational complexity, I have added relevant indicators to measure this problem.
General Comments:
The authors should do comparison not only with baselines listed in the manuscript but also with some light-weight models, for example, RUAS(2021).
Response:
Thanks for this valuable suggestion. When this algorithm is compared with other algorithms, I ' ve already considered this problem, and Zero-DCE + + is a lightweight algorithm.
General Comments:
Moreover, the list of references used in this manuscript is not up-to-date one, whose the years of publish are mostly before 2021. There are recently (2022-2023) works have quite similar objectives.
Response:
Thank you for your advice. In response to this problem, I added nearly two years of literature to supplement.
General Comments:
The authors should put extra care for writing manuscript. In fact, there are repeated parts. For example, from line 36-47, the authors repeatedly mentioned how existing method ignore noise and color imbalance. Or the beginning character of a sentence was not capitalized. For example, line 32. Or no space between two sentence. For example, line 35.
Response:
Thank you for your advice and we apologize for the grammatical errors in the paper. In response to these problems, I have made changes in the paper. Thank you for reviewing this article. (Line36-42, this paragraph has been rephrased, and other places have also made changes.)

Reviewer 3 Report
The manuscript presents an image low-illumination restoration framework. The authors propose 3 connected deep neural networks to gradually decompose the image into illumination and reflection component, fuse them to get a noise-suppressed reflectance map, and enhance this map to get a restored image. The experiments section show qualitative and quantitative improvements compared to other methods.
The method shows good promise. However, there are a few concerns that need to be addressed by the authors.
1) DRBN paper shows a PSNR of 20.13 while a PSNR of 15.141 is reported in this manuscript. Also, DRBN was compared against a number of methods in the LOL dataset, and a lot of them showed better PSNR than the proposed method, in the original DRBN method. Hence, it raises a concern if the authors put worse average results compared to most literature, in order to show their benefit in the current manuscript.
Same goes for RRDNet on DICM, LIME, MEF, and NPE datasets using NIQE score. Scores do not match and there are other methods reporting better scores in those datasets.
What is the reason of this discrepancy?
2) Since the method is supposed to be end-to-end, why not merge the ReconNet and EnhanceNet part together? In practice, both of these networks virtually operate on the R_low and I_low inputs. So, they can be used in a single block.
Also, it is quite understood that ReconNet cannot do the job itself, since the output needs to be enhanced anyway. So, instead of using 3 blocks with different losses, which makes training difficult, why not put the two blocks together?
The justification of two different networks is not clear from the manuscript. The ablation study does not cover this part as well.
3) From all the quantitative results, ignoring any results discrepancy (see concern 1), the proposed method has similar results to SOTA methods (sometimes better and sometimes worse). Specifically, LOE scores are mostly comparable. In the light of this fact, the 3rd conclusion by the authors of outperforming existing techniques by significant margin is incorrect. If the authors would like to highlight specific evidence to support this claim, please do so. Otherwise, the authors are advised to rephrase this claim.
Author Response
Response to Reviewer 3
General Comments:
The manuscript presents an image low-illumination restoration framework. The authors propose 3 connected deep neural networks to gradually decompose the image into illumination and reflection component, fuse them to get a noise-suppressed reflectance map, and enhance this map to get a restored image. The experiments section show qualitative and quantitative improvements compared to other methods.
The method shows good promise. However, there are a few concerns that need to be addressed by the authors.
Response:
We really appreciate your comprehensive comments and constructive suggestions. We have benefited greatly from your suggestions. We read and consider each comment very carefully, and thoroughly revise the manuscript according to your comments and suggestions. We hope that the manuscript reads more convincingly after the revision.
Comment 1:
DRBN paper shows a PSNR of 20.13 while a PSNR of 15.141 is reported in this manuscript. Also, DRBN was compared against a number of methods in the LOL dataset, and a lot of them showed better PSNR than the proposed method, in the original DRBN method. Hence, it raises a concern if the authors put worse average results compared to most literature, in order to show their benefit in the current manuscript.
Response:
Thank you for your careful review and valuable suggestion. I think the reason for this difference is due to the different selection of pictures in the test set and the difference in the training set selected by some networks during training.In the article Low-Light Image and Video Enhancement Using Deep Learning : A Survey (https://arxiv.org/pdf/2104.10729.pdf).The author 's test results of PSNR and SSIM evaluation indicators in the LOL test set are not much different from my test results.
Comment 2:
Same goes for RRDNet on DICM, LIME, MEF, and NPE datasets using NIQE score. Scores do not match and there are other methods reporting better scores in those datasets.
What is the reason of this discrepancy?
Response:
Thank you for your advice. The reason for the inconsistent NIQE scores may be that the image selection in the test set is more biased towards images with lower brightness and more complex brightness background distribution. The algorithm introduced in this paper incorporates an attention module into the Enhance-Net network, enhancing its capability to efficiently handle variations in brightness within the background. Therefore, the algorithm achieves good results on the index. In response to this problem, I re-done the relevant experiments of NIQE and LOE indicators, and comprehensively selected the images of the test set for experiments. And made changes in the paper.
Comment 3:
Since the method is supposed to be end-to-end, why not merge the ReconNet and EnhanceNet part together? In practice, both of these networks virtually operate on the R_low and I_low inputs. So, they can be used in a single block.Also, it is quite understood that ReconNet cannot do the job itself, since the output needs to be enhanced anyway. So, instead of using 3 blocks with different losses, which makes training difficult, why not put the two blocks together? The justification of two different networks is not clear from the manuscript. The ablation study does not cover this part as well.
Response:
Thank you for your suggestion. Can not be merged because of the different functions of the two networks. Recon-Net is used to denoise and suppress color fading. Ehance-Net is used for brightness enhancement. The loss function of these two parts also plays a different role. And the final output results of the two networks are also inconsistent. If it is combined into one for training, it will lead to poor enhancement results. I added this in the ablation experiment.
Comment 4:
From all the quantitative results, ignoring any results discrepancy (see concern 1), the proposed method has similar results to SOTA methods (sometimes better and sometimes worse). Specifically, LOE scores are mostly comparable. In the light of this fact, the 3rd conclusion by the authors of outperforming existing techniques by significant margin is incorrect. If the authors would like to highlight specific evidence to support this claim, please do so. Otherwise, the authors are advised to rephrase this claim.
Response:
Thank you for your careful review and valuable suggestion! For this point, I after careful consideration, really express inappropriate in the article. Therefore, I restate it in the paper. Thank you for reviewing this article.

Round 2
Reviewer 1 Report
Although the authors have answered my comments, the paper has not been substantially changed. I believe that an article should be self-contained and therefore clarifications should be present in the article, not in the answers given to the reviewer.
Moreover, from my point of view, it is still not clear which are the inputs of the initial part of the neural network: How or where are the S_nor and S_low images obtained?
On the other hand, the authors have indicated that some answers are found in the article Deep Retinex Decomposition for Low-Light Enhancement (https://arxiv.org/pdf/1808.04560.pdf) and, reviewing it, I think that the proposal made does not have major changes with the indicated article, therefore, it should not be published.
Author Response
Comment 1:
Although the authors have answered my comments, the paper has not been substantially changed. I believe that an article should be self-contained and therefore clarifications should be present in the article, not in the answers given to the reviewer.
Response:
Thank you for valuable advice. I 'm sorry that I only replied to your letter and did not make enough changes in the paper. In response to these questions you raised, I made a reflection and made changes in the relevant places.
Comment 2:
Moreover, from my point of view, it is still not clear which are the inputs of the initial part of the neural network: How or where are the S_nor and S_low images obtained?
Response:
Thank you for your correction. The network can be divided into three parts. As the initial input part of the network, Decom-Net is used to decompose the image into illumination component and reflection component. In the training phase, the pictures Snor and Slow in the paired dataset LOL need to be input into Decom-Net at the same time, and the training effect is achieved through parameter sharing. Snor and Slow represent the situation of an image under normal lighting and low brightness environments. They come from the paired dataset LOL and are read in Decom-Net, which is decomposed into illumination component and reflection component.
Comment 3:
On the other hand, the authors have indicated that some answers are found in the article Deep Retinex Decomposition for Low-Light Enhancement (https://arxiv.org/pdf/1808.04560.pdf) and, reviewing it, I think that the proposal made does not have major changes with the indicated article, therefore, it should not be published.
Response:
Thank you for reviewing my answers. I 'm sorry that my previous answer didn 't satisfy you, then I revised your previous question on the paper. Thank you for your correction of this article.
Reviewer 2 Report
Thank you for revising the manuscript. However, the novelty and feasibility of this study are big concerns. Previously, reviewer commented that the list of references is not up-to-date. It does not mean that the authors just simply add newer references. It requires the authors to have more up-to-date literature review and comparison. Moreover, the authors claimed that Zero-DCE ++ was used as a lightweight baseline, instead of RUAS(2021). However, it was proven that Zero-DCE ++ underperform in comparison with RUAS in terms of PSNR, MAE, SSIM, etc. So comparing with Zero-DCE++ is not sufficient. In addition, there exists studies that have extremely similar approaching manner, for example URetinex (CVPR 2022). Particularly, URetinex can also deal with the problem of color distortion (through its initialization module), noise, loss of details. The comparison with such methods are required to prove the feasibility of this study.
Author Response
General Comments:
Thank you for revising the manuscript. However, the novelty and feasibility of this study are big concerns.
Response:
We really appreciate your comprehensive comments and constructive suggestions. We have benefited greatly from your suggestions. We read and consider each comment very carefully, and thoroughly revise the manuscript according to your comments and suggestions. We hope that the manuscript reads more convincingly after the revision.
Comment 1:
It does not mean that the authors just simply add newer references. It requires the authors to have more up-to-date literature review and comparison.
Response:
Thank you for your advice. I am sorry for the misunderstanding of your previous suggestions. The author reviews and compares the newly added methods and modifies them in the article.
Comment 2:
Moreover, the authors claimed that Zero-DCE ++ was used as a lightweight baseline, instead of RUAS(2021). However, it was proven that Zero-DCE ++ underperform in comparison with RUAS in terms of PSNR, MAE, SSIM, etc. So comparing with Zero-DCE++ is not sufficient.
Response:
Thank you for your valuable suggestions regarding this article. Upon further reflection, I realized that I did not thoroughly consider the limitations of Zero-DCE++ in relation to the indicators during my comparison with the lightweight algorithm. As a result, I have included a comparison of the RUAS algorithm in the evaluation criteria and subsequently made appropriate modifications to the paper.
Comment 3:
In addition, there exists studies that have extremely similar approaching manner, for example URetinex (CVPR 2022). Particularly, URetinex can also deal with the problem of color distortion (through its initialization module), noise, loss of details. The comparison with such methods are required to prove the feasibility of this study.
Response:
Thank you for your suggestion. The previous experimental comparison did lack relevant considerations, so I added the SCL-LLE ( AAAI 2022 ) method for comparison. This method takes into account factors such as detail enhancement, color consistency and light distribution, which is suitable for comparison with the method in this paper. Thank you for your review of this article.
